# Differential Regulation of POC5 by ERα in Human Normal and Scoliotic Cells

**DOI:** 10.3390/genes14051111

**Published:** 2023-05-19

**Authors:** Amani Hassan, Edward T. Bagu, Shunmoogum A. Patten, Sirinart Molidperee, Stefan Parent, Soraya Barchi, Isabelle Villemure, André Tremblay, Florina Moldovan

**Affiliations:** 1Research Center CHU Sainte-Justine, 3175 Chemin de la Cote-Sainte-Catherine, Montréal, QC H3T 1C5, Canadas.molidperee@gmail.com (S.M.); stefan.parent@umontreal.ca (S.P.); soraya.barchi.hsj@ssss.gouv.qc.ca (S.B.); andre.tremblay.1@umontreal.ca (A.T.); 2Department of Basic Biomedical Sciences, Sanford Medical School, University of South Dakota, Vermillion, SD 57069, USA; edward.bagu@usd.edu; 3INRS Center Armand-Frappier Santé Biotechnologie, 531 Boul des Prairies, Laval, QC H7V 1B7, Canada; kessen.patten@iaf.inrs.ca; 4Department of Mechanical Engineering, Polytechnique Montréal, 2500 Chemin de Polytechnique, Montréal, QC H3T 1J4, Canada; isabelle.villemure@polymtl.ca; 5Department of Obstetrics & Gynecology, Université de Montréal, Montréal, QC H3T 1J4, Canada; 6Department of Biochemistry and Molecular Medicine, Université de Montréal, Montréal, QC H3C 3J7, Canada; 7Centre de Recherche en Reproduction et Fertilité, Université de Montréal, Saint-Hyacinthe, QC J2S 2M2, Canada; 8Department of Stomatology, Faculty of Dentistry, Université de Montréal, 2900 Edouard Monpetit Boulevard, Montréal, QC H3T 1J4, Canada

**Keywords:** puberty, POC5, 17β-estradiol, estrogen, E2, osteoblasts, control NOBs, mutant *POC5*^A429V^ AIS osteoblasts, estrogen resistance, estrogen receptor

## Abstract

Adolescent idiopathic scoliosis (AIS) is a complex three-dimensional spinal deformity. The incidence of AIS in females is 8.4 times higher than in males. Several hypotheses on the role of estrogen have been postulated for the progression of AIS. Recently, Centriolar protein gene POC5 (*POC5*) was identified as a causative gene of AIS. POC5 is a centriolar protein that is important for cell cycle progression and centriole elongation. However, the hormonal regulation of POC5 remains to be determined. Here, we identify *POC5* as an estrogen-responsive gene under the regulation of estrogen receptor ERα in normal osteoblasts (NOBs) and other ERα-positive cells. Using promoter activity, gene, and protein expression assays, we found that the *POC5* gene was upregulated by the treatment of osteoblasts with estradiol (E2) through direct genomic signaling. We observed different effects of E2 in NOBs and mutant *POC5*^A429V^ AIS osteoblasts. Using promoter assays, we identified an estrogen response element (ERE) in the proximal promoter of *POC5*, which conferred estrogen responsiveness through ERα. The recruitment of ERα to the ERE of the *POC5* promoter was also potentiated by estrogen. Collectively, these findings suggest that estrogen is an etiological factor in scoliosis through the deregulation of POC5.

## 1. Introduction

Scoliosis is a complex three-dimensional deformity of the spine with unclear etiology, with a prevalence of 1.5 to 3% in the general population [1]. The most common form of scoliosis is idiopathic scoliosis (IS), which can be further classified according to the age of onset, as infantile, juvenile, and adolescent [1]. Adolescent idiopathic scoliosis (AIS) principally affects children older than 10 years during the course of skeletal maturation [2]. AIS is more prevalent in girls as compared to boys and tends to be associated with more severe clinical deformities [1]. At present, there is no consensus on the aetiology of AIS, although it is accepted that it is due to a combination of multiple factors that have been grouped as biomechanical, genetic, and metabolic factors [3,4,5,6]. 

In young females with scoliosis, there is a decrease in bone density that coincides with the period when estrogen signaling is activated [3,4,5]. Estrogen is the key endocrine contributor to growth, the attainment of puberty, and development in females [7,8,9,10]. Interestingly, patients with AIS were reported to harbor estrogen receptor(ER) isoforms and polymorphism. It was suggested that the aberrant estrogen signaling in AIS patients was associated with the presence of these polymorphisms affecting both *ERα* and *ERβ* and the G-protein coupled estrogen receptor 1 (GPER) [11,12,13,14,15,16,17,18,19,20]. The coincidence of such hormonal disrupted signaling events with bone abnormalities led to the suggestion that AIS is an endocrine disease with estrogen deregulation as the main etiological factor [13]. At present, it is not clear how estrogen is involved in the pathogenesis of AIS; however, it is possible that estrogen may interact with other pathophysiological factors such as melatonin, growth, biomechanical stress, and metabolic stress that are believed to influence the development of scoliosis [3]. 

During puberty in girls, there is a rise in estrogen levels that is associated with the reduction in bone turnover markers [21]. This process subsequently causes the closure of the epiphyseal growth plates by decreasing periosteal apposition and endosteal resorption within cortical bone, as well as bone remodeling within cortical and cancellous bone [22]. These effects of estrogen on the bone are promoted through the induction of the apoptosis of chondrocytes in the growth plate and of osteoclasts within the cortical and cancellous bone. Based on the established effects of estrogen on bone formation, growth, maturation, and turnover, it is expected that estrogen is a contributing factor in the development and progression of scoliosis [22]. 

In a study of a multiplex family, three mutations, c.G1336A (p.A446T), c.G1363C (p.A455P), and c.C1286T (p.A429V), were identified in the protein POC5 in patients with IS, thereby suggesting that *POC5* is a causative gene of AIS [23]. The relationship between POC5 and AIS was further corroborated in a case-control study in Chinese patients, showing that the common variant rs6892146 is associated with the development of AIS in the Chinese population [24]. POC5 is a centrin-binding protein that is required for assembling the centriole and cellular proliferation. It is localized at the mother and daughter centrioles throughout the cell cycle. POC5 is a conserved protein encoded by the *POC5* gene located on chromosome 5q13.3 in humans [25,26]. 

In this work, we identified *POC5* as an estrogen-responsive gene in osteoblasts and other ERα-positive cells. We analyzed the gene and protein levels of POC5, its cellular localization and multi centriole formation, and identified estrogen response elements (EREs) in the *POC5* promoter under the control of ERα. Interestingly, we also show a reduced response to estrogen in AIS human osteoblasts carrying a *POC5*^A429V^ mutation compared to control normal osteoblasts (NOBs). Our findings suggest that the aberrant estrogen signaling in osteoblasts is a driving factor of the development and progression of spinal deformity, thereby providing essential insights into the importance of the contribution of estrogen to AIS.

## 2. Materials and Methods

### 2.1. In-Silico Analysis of Gene Expression

*POC5* microarray expression in normal human tissues was extracted from the Gene Expression Omnibus (GEO) database available (GDS3834/10890) from US National Center for Biotechnology Information (NCBI). The profiles of the target genes in human tissues were collected from different studies, and the mean transcript levels were determined. Data are presented as the mean ± SD.

### 2.2. Cells and Treatments

Human mutant *POC5*^A429V^ AIS osteoblasts and control NOB cells were extracted from tissues and were collected with the consent of patients after the approval by the Institutional Ethics Committee Board of CHU Sainte-Justine, Montreal, Canada. Huh-7 cells, a hepatoma-derived immortalized cell line of tumorigenic epithelial-like cells, were cultured as previously published [27]. MCF-7, a breast cancer cell line, was purchased from the American Type Culture Collection (ATCC, Manassas, VA, USA) and cultured as recommended by the vendor (ATCC). The 17β-estradiol (E2) and 4-hydroxy-Tamoxifen (4-OHT) were purchased from Sigma–Aldrich (Oakville, ON, Canada). Fulvestrant (ICI-182, 780) was purchased from TOCRIS (Minneapolis, MN, USA) and was separately reconstituted with 100% ethanol into solutions of a 2 × 10^−2^ M stock concentration and then stored at −20 °C until use. 

During the evaluation of the effect of 17β-estradiol, 4-hydroxy-Tamoxifen (4-OHT), and Fulvestrant (ICI-182, 780) on the expression of POC5 in Huh-7, MCF-7, and osteoblasts, the cells used were cultured in their respective media, devoid of phenol red, containing 10% FBS that was previously stripped with charcoal. 

### 2.3. Samples, Osteoblast Isolation, and Culture

Bone biopsies were collected during surgery from the vertebrae of AIS patients (varying from T3 to L4) and from non-scoliotic patients undergoing surgery for a traumatic spinal condition [28] (Table 1). Bone fragments were reduced to smaller pieces with a bone cutter under sterile conditions as previously described [29]. The fragments were incubated at 37 °C in 5% CO_2_ in a 100 mm culture dish in the presence of α modified Eagle medium containing 10% fetal bovine serum (FBS) (Wisent Bioproducts) and 1% penicillin and streptomycin (Invitrogen, Burlington, ON, Canada). After a 28-day period, the osteoblasts derived from the bone pieces were separated at confluence from the remaining bone fragments by trypsinization. To confirm the osteoblast phenotype and mature osteoblast cell function, alizarin red staining was performed. 

### 2.4. Alizarin Red Staining

The mineralization rate of osteoblasts, derived from non-scoliotic patients (normal osteoblasts, NOBs) and mutant *POC5*^A429V^ AIS osteoblasts, was determined by alizarin staining. Osteoblasts were cultured until confluency (2 × 10^5^) in 24-well plates using α modified Eagle medium (Wisent Bioproducts, St-Bruno, QC, Canada) containing ascorbic acid 50 µg/mL, dexamethasone 10 nM, and glycerophosphate 2.5 mM. The mineralization rate was determined after cell fixation with PFA 10% and alizarin red staining (40 mM, pH = 4.1). Alizarin red quantification was performed using a microplate reader at an absorbance of 405 nm. For the standard curve, serial dilutions were prepared (0–500 μg/mL).

### 2.5. Alkaline Phosphatase (ALP) Staining

Osteoblasts were cultured in DMEM/F12 until confluency (2 × 10^5^ cells). Three days before alkaline phosphatase staining, the medium was replaced with MEM supplied with 10% FBS, 1% PS in addition to ascorbic acid 50 µg/µL, dexamethasone 10 nM, and β-glycerophosphate 2.5 mM. After 72 h, cells were washed with PBS and then fixed with 3.7% PFA. The cells were then washed with PBS 1X and incubated with a solution of 1.2 mg of naphthol AS-MX phosphate (Sigma, Toronto, ON, Canada), 2.5 mg of fast red TR (Sigma, ON, Canada), and 62.5 μL ethylene glycol monoethylether in 6 mL of TBS 0.1 M, pH 9.5 (Tris 0.1 M, NaCl 0.15 M, pH 9.5) for 1 h at room temperature (RT). The cells were then washed with water and observed under a light microscope.

### 2.6. RNA Isolation, Reverse Transcription, PCR, and Real-Time PCR

Total RNA from Huh-7 (Hepatocytes), MCF-7 (breast cancer cells), and osteoblasts was isolated using TRizol as recommended by the manufacturer (Invitrogen, Canada). The RNA (2 μg) was used as a template to synthesize the first-strand cDNA using iScript reverse transcriptase from Bio-Rad (Mississauga, ON, Canada). The quantification of gene expression was performed by a 7900HT Fast Real-Time PCR System (Thermo Fisher Scientific, Toronto, ON, Canada) with iQ™ SYBR^®^ Green Supermix. The oligonucleotides used are listed in Table 2. The fold change was calculated using the delta CT method.

### 2.7. Plasmid Construction

Two promoter fragments, at positions −3653/−1561 and −1481/+48 base pairs from the transcriptional start sites, upstream of the 5′-flanking end of the *POC5* un-translated region, were generated by PCR using the primers listed in Table 1 and genomic DNA that was isolated from the Huh-7 cell lines. The fragments were then cloned into separate pGL3 basic luciferase reporter plasmid vectors (Promega, Madison, WI, USA) after restriction digestion with Nhe I/Bgl II and Nhe I/Xho I. Deletion constructs of the 1529 bp fragments −555/+48 and −248/+48 were generated using the primers listed in Table 3 with the *POC5* plasmid −1481/+48 as a template. All plasmids were verified by digestion with restriction enzymes and sequencing (McGill University and Génome Québec Innovation Centre). The expression vector encoding full-length wild-type human ERalpha (pEGFP-hERα) was acquired from Addgene (#28230).

### 2.8. In-Silico Analysis of the POC5 Promoter

Putative response elements (REs) in the *POC5* promoter for the estrogen receptor α (*ERα*) were determined by the multiple bio-informatics tool [http://biogrid-lasagna.engr.uconn.edu/lasagna_search/] (accessed on 1 February 2013)in the −3653/−1561 fragment of the *POC5* promoter covering 3 estrogen response element (ERE) sites: 1 (−3638 /−3541), 2 (−3407/−3400), and 3 (−1845/−1838), and the −1481/+48 fragment has 3 ERE sites 1 (−1012/−1005), 2 (−845/−837), and 3 (−755/−747).

### 2.9. Protein Lysate Preparation and Western Blotting

Whole-cell protein lysates were prepared from cell lines using RIPA buffer from Pierce Thermo Fisher Scientific (Toronto, ON, Canada, cat. 89900) (25 mM Tris•HCl pH 7.6, 150 mM NaCl, 1% NP-40, 1% sodium deoxycholate, 0.1% SDS) supplemented with protease and phosphatase inhibitors (Roche Diagnostics, Mannheim, Germany). To perform the Western blot, equal amounts of protein were resolved using SDS/polyacrylamide gel electrophoresis. Afterwards, proteins were transferred onto a nitrocellulose membrane and blocked in phosphate-buffered saline containing 0.05% Tween-20 and 20% skim milk powder. Membranes were incubated with primary antibody (POC5, Abcam, Torronto, ON, Canada cat ab250957), collagen 1A2 (Santa Cruz Biotechnology, Inc. Dallas, TX USA, cat sc-166865), and β-actin (Santa Cruz, cat sc-47778) overnight at 4 °C and then washed with phosphate-buffered saline Tween-20. Afterwards, membranes were incubated with a secondary antibody conjugated with horseradish peroxidase (Anti-rabbit, Thermo Fisher Scientific, Toronto, ON, Canada, cat 31462) for 1 h at RT. After incubation, proteins were visualized by enhanced chemo-luminescence.

### 2.10. Transient Transfection Assays

Transfections of Huh-7 hepatoma cells were performed in 24-well plates using the Lipofectamine^TM^ 2000 (Invitrogen, Burlington, ON, Canada) as recommended by the manufacturer. When cells attained 80% confluence, they were co-transfected with 990 ng/well of the different *poc5* promoter constructs (−1481, −552, −248) along with 10 ng of phRL-TK (Renilla Luciferase), Promega, Madison, WI, USA). The total DNA per well was kept at 750 ng/well in the 24-well plates by co-transfection with the empty expression vector pcDNA3. During the evaluation of the effect of ERα on *hPOC5* in the presence or absence of 17β-estradiol, ERα-negative Huh-7 cells were co-transfected with different *poc5* promoter constructs along with either the expression vector encoding the full-length human ERα-protein (25 ng/well) or the empty pcDNA-3 vector (Invitrogen, Burlington, ON, Canada). Five hours following transfection, the media in which cells were cultured were replaced with serum-free media. After 12 h, cells were then cultured in fresh serum-free media with or without one of the following treatments, E2, 4-OHT or ICI-182, 780, for 24 h. In order to evaluate the basal luciferase activities for each construct, controls for each full-length promoter construct were co-transfected with an empty pcDNA-3 vector (Invitrogen, Burlington, ON, Canada) and then cultured in the vehicle. In all experiments, the data reported represent the average of at least three experiments, performed in triplicate, using at least three different DNA preparations.

### 2.11. Immunofluorescence

Control NOBs and mutant *POC5*^A429V^ AIS osteoblasts were cultured in vitro to attach to the Labtek (NUNC, Thermoscientific) overnight. On the second day, cells were fixed in 70% ethanol/0.1% triton on ice for 30 minutes (min). Cells were then washed with PBS and permeabilized with 0.1% Triton in PBS for 15 min. Cells were washed once with 0.5% BSA in PBS/Triton (PBT) and then blocked with 2% BSA in PBT for 45 min. Cells were washed after that and incubated with the anti-POC5 (Abcam, cat ab250957) and anti-centrin antibodies (LifeSpan Biosciences, cat LS C482434, Carlsbad, CA, USA) at (1/200) ON at 4 °C with gentle shaking. Cells were then incubated with Alexa Flour 555 (Thermo Fisher Scientific Toronto, ON, Canada cat A21422) and Alexa Flour 488 (Life Technologies USA, cat A11008) for one hour. Cells were mounted and subjected to nuclear staining at the same time using Prolong Gold antifade reagent with DAPI (Life Technologies). Immunostaining was observed at magnification ×40.

### 2.12. Chromatin Immunoprecipitation (ChIP)

ChIP was performed as described [30]. Briefly, MCF-7-ERα cells were cultured in phenol red-free medium and 10% charcoal-stripped FBS and then treated with 10^−7^ M E2 or vehicle for 1 h. After fixation with 2% formaldehyde, cells were lysed, and the precleared chromatin supernatants were immunoprecipitated with the respective antibodies: specific anti-ERα (Santa-Cruz, cat Sc-542) or non-specific IgG at 4 °C. Bound DNA was purified with phenol/chloroform and used as a template for subsequent amplification using primers (Table 4) that encompassed the respective specific binding elements within the proximal *POC5* promoter region. Fold enrichment values were calculated using the Ct value of each ChIP sample compared to the Ct value of input DNA.

### 2.13. Thymidine Incorporation Assay

Cells were grown to subconfluence in complete medium at 37 °C in a humidified atmosphere with 5% CO_2_ in 162 cm^2^ flasks. Cells were passaged 1:10. Exponentially growing cells exhibited the best uptake. Cells were trypsinized, and cell density was estimated in the resulting suspension. The suspension was diluted in complete medium to the selected seeding density and dispensed (40 μL/well). Cells were incubated overnight at 37 °C, 5% CO_2_. On the second day, media were removed from the wells and the cells were washed with 50 μL sterile PBS/well. Then, [methyl-14C] thymidine was diluted to 0.625 μCi/mL in assay medium. Subsequently, 80 μL [methyl-14C] thymidine was added to the wells to yield a final concentration of 0.05 μCi/mL. Then, [methyl-14C] thymidine was added to medium-only solution to establish control wells (background counts). Plates were covered with a clear plastic plate seal and counted immediately at appropriate intervals every 24, 48, and 72 h. Cells were regularly checked under the microscope for growth and morphology.

### 2.14. Statistical Analysis

The statistical significance of the results was determined by Student’s *t*-test and one-way analysis of variance (ANOVA) as indicated for each experiment. The luciferase and qPCR experiments were performed in triplicate three times (n = 3). The Western blot experiments were performed three times; *p* < 0.05 was accepted as statistically significant. 

## 3. Results

### 3.1. Gene Expression Profile of POC5 in Multiple Human Tissues 

Little is known about the tissue distribution of POC5. Thus, we first analyzed the expression levels of *POC5* in different human tissues (Figure 1) using the Gene Expression Omnibus (GEO) database at the US National Center for Biotechnology Information (NCBI). *POC5* expression shows a wide distribution in different tissues with the highest levels observed in the pancreas, heart, lung, intestine, brain, and colon. Very low levels are observed in the skin and adipose tissue. Interestingly, *POC5* is expressed at high levels in the bone. This proposes a wider function than expected and an undetermined role of *POC5* in bone.

### 3.2. The POC5^A429V^ AIS Osteoblasts Have a Reduced Mineralization Rate Compared to NOB Cells

To investigate the impact of the A429V mutation and the role of *POC5* in the differentiation and function of primary osteoblasts derived from healthy donors without scoliosis and AIS patients, we performed alizarin red staining, alkaline phosphatase, and Western blot analyses of COL1A2 (Figure 2 and Appendix A). Our results show that AIS cells had a significantly lower mineralization rate compared to normal cells, indicating a detrimental effect of the A429V mutation.

### 3.3. Centriolar Protein POC5 Is Expressed at Higher Levels in NOBs Compared to Mutant POC5^A429V^ AIS Osteoblasts 

First, we determined the expression levels of POC5 (gene and protein) in control (NOBs) and mutant *POC5*^A429V^ AIS osteoblasts. At the gene level, *POC5* was highly expressed in control NOB cells, while the expression was significantly downregulated in mutant *POC5*^A429V^ AIS osteoblasts (*p* < 0.001) (Figure 3A). Likewise, a significant reduction (*p* < 0.001) in POC5 protein levels was observed in mutant *POC5*^A429V^ AIS cells compared to NOBs (Figure 3B,C). 

### 3.4. Estrogen Upregulates POC5 Expression in a Dose-Dependent Manner in Control NOBs but Not in Mutant POC5^A429V^ AIS Osteoblasts

We next examined the regulation of mRNA expression of *POC5* in response to E2 at different time points in control NOBs and mutant *POC5*^A429V^ AIS osteoblasts. NOB cells had a maximal response to E2 after 6 h of treatment, and the response dramatically decreased at 12 h and 24 h in both cell types, which suggests that *POC5* is an early responsive gene to E2 (Figure 4A,C). *POC5* gene expression analysis showed a significant induction of *POC5* by E2 in NOBs after 6 h of treatment (5-fold) (Figure 4A). However, there was no significant induction in AIS cells at the same time point (Figure 4C). POC5 protein levels, in response to different doses of E2, were tested in control NOBs and in mutant *POC5*^A429V^ AIS osteoblasts. E2 upregulated POC5 expression in control NOBs starting at 10^−7^ M E2 and reached a significant maximal response at 10^−9^ M (Figure 4B). In mutant *POC5*^A429V^ AIS osteoblasts, the difference between control and E2-treated samples was not significantly different (Figure 4D). To determine if there was any effect of the E2 treatment on the subcellular localization of POC5 and if there were different effects on control NOBs and mutant *POC5*^A429V^ AIS osteoblasts, we performed immunofluorescence tests on both cells after treatment with E2. We found an increase in the centriolar POC5 protein content in response to estrogen as observed by the enhanced staining of POC5 and linear centriolar structures in both cells, as confirmed by centrin2 staining (Figure 4E,F). However, there was incomplete colocalization of these structures with centrin in mutant *POC5*^A429V^ AIS osteoblasts.

### 3.5. E2 Upregulates POC5 Gene and Protein Expression Levels in MCF-7 and Huh-7 Cells

In order to ascertain the estrogen regulation of POC5, we next analyzed the POC5 response in other ERα-positive cells, such as highly estrogen-responsive MCF-7 and Huh-7 cells. *POC5* gene expression was strongly induced at 3 h of E2 treatment (12-fold versus the control) in MCF-7 cells, while longer treatment periods showed reduced activation levels indicative of a rapid response to estrogen (Figure 5A). POC5 protein levels were also increased at 3 h of E2 treatment (Figure 5B). Similarly, *POC5* expression levels were induced at 3 h of E2 treatment in Huh-7 cells, whereas a negative regulation was observed for longer treatments (Figure 5C). POC5 protein levels were also significantly induced in these cells (Figure 5D). These findings indicate that the estrogen regulation of POC5 expression might extend to other ERα-positive cells and occurs in a rapid fashion.

### 3.6. ERα Antagonists Attenuate the E2-Induced Regulation of POC5

To determine if the regulation of *POC5* by E2 occurred through direct regulation by the estrogen receptor ERα, we tested whether ER antagonists Fulvestrant (ICI-182, 780) and 4-hydroxy tamoxifen (OHT) were effective in impeding the estrogen induction of *POC5*. ER-positive MCF-7 cells were treated with antagonists in the presence and absence of E2. At the mRNA level, *POC5* gene expression was abolished by ICI-182, 780 treatment at similar levels to the control. *POC5* was downregulated after exposure to ICI-182, 780 and E2 compared to E2-treated cells (Figure 6A). At the protein level, both antagonists significantly abolished the E2-induced upregulation of POC5 (Figure 6B). 

### 3.7. The Proximal Promoter Region of POC5 Confers Estrogen Responsiveness through ERα

Based on the sequence analysis of the *POC5* promoter, we identified that the first 1481 bp region respective to the transcriptional start site contains three putative estrogen response elements (EREs) located at positions −455/−432, −410/−389, and −291/−272 (Figure 7A). In order to determine whether this cluster of estrogen response elements was functional in translating estrogen responsiveness to *POC5,* we generated various promoter constructs of the *POC5* gene and performed luciferase reporter gene assay. A deletion construct missing the three EREs (position −1561 to −3653) was not significant in conferring estrogen responsiveness, whereas a fragment consisting of the three EREs (position −1481 to +48) contributed 2.3-fold of the response to estrogen (Figure 7B). Similarly, a 2.5-fold increase was also observed with the −552/+48 fragment. However, such response was lost using the −248/+48 fragment of the promoter missing all three EREs. This suggests that the ERE-containing −552/+48 fragment is important for the induction of the estrogen response of *POC5*.

### 3.8. Estrogen-Dependent Recruitment of ERα to the POC5 Proximal Promoter

We next determined the recruitment of ERα onto the *POC5* gene promoter in response to estrogen using chromatin immunoprecipitation (ChIP) assay. Using primers spanning the ERE1/2-containing −478 to −423 region and the ERE3-containing −291 to −272 region, we found a strong recruitment of ~300-fold and ~230-fold of ERα, respectively, to the *POC5* gene in response to estrogen compared to untreated cells (Figure 8A,B). This indicates that the proximal promoter region containing the three EREs is involved in mediating the estrogenic response of the *POC5* gene.

### 3.9. E2 Upregulates Bone Markers and ER Isoforms in Control NOBs but Not in Mutant POC5^A429V^ AIS Osteoblasts

Given the role that E2 plays in the growth and maturation of bone, we next sought to determine if there were differential effects of E2 on bone markers in control NOBs and mutant *POC5*^A429V^ AIS osteoblasts. We investigated the expression of bone markers in response to E2. Alkaline phosphatase (*ALPL*), *RUNX2,* and *SPP1* (osteopontin, secretes phosphoprotein 1) were significantly upregulated while osteocalcin (bone γ-carboxyglutamic acid-containing protein (*BGLAP*)), was significantly (*p* < 0.01) downregulated in control NOBs (*p* < 0.05). In mutant *POC5*^A429V^ AIS osteoblasts, there was no significant change in all tested bone markers in response to E2 (Figure 9A). We also determined the regulation of *ERα* and *ERβ* by E2. ERα expression was highly upregulated in control NOBs and mutant *POC5*^A429V^ AIS osteoblasts (*p* < 0.05 and *p* < 0.01). *ERβ* was downregulated by E2 in control NOB cells (*p* < 0.01), and there was no significant change in mutant *POC5*^A429V^ AIS (Figure 9B). We also compared the proliferation rate of NOBs and mutant *POC5*^A429V^ AIS osteoblasts by thymidine incorporation assay (Appendix A). NOB cells had a higher proliferation rate at different tested time points.

## 4. Discussion

Since the advent of high-throughput sequencing technologies and the global advance in knowledge of the genome, several genes have been associated with AIS, but for the most part, without an explicit molecular mechanism [31]. Among the genes that have been confirmed to be involved in the development of scoliosis in humans (*TBX6*, *FGF3*, *LBX1*, *POC*5, *TTLL1*) [23,24,32,33,34,35] and vertebrates (*Kif6* and *Ptk*) [36,37], several are ciliary genes [23,32,33,34,35,36,37]. The ciliary-related gene *POC5* explains 10–14% of AIS family cases [23,24,25,26,27,29,30,31,32,33,34,35,36,37,38]. One of the most important questions asked over decades is why girls are more affected than boys and why puberty is the period during which AIS predominantly progresses in girls. The novel finding of our study is that we show for the first time that the centriolar protein POC5 is upregulated by estrogen through a direct effect of the estrogen receptor ERα. We demonstrated that the regulation of POC5 is through direct genomic signaling where the estrogen receptor directly binds to the estrogen response elements located within the −552/+48 fragment of the *POC5* promoter. To study the responsiveness of the POC5 gene and protein to E2, we used human osteoblasts. To confirm a wider response of POC5 to estrogen, we expanded this analysis to other ERα-positive cell lines, such as MCF-7 and Huh-7 cells. In all tested cells, POC5 was upregulated upon short exposure of cells to estrogen, suggesting that *POC5* is an early-responsive gene. Interestingly, mutant *POC5*^A429V^ AIS osteoblasts exhibited a reduced response to E2 compared to control NOB cells, suggesting a possible mechanism for estrogen resistance [28]. At the protein level, there was a dose-response to E2 in control NOB cells and no significant change in mutant *POC5*^A429V^ AIS osteoblasts. Bone formation markers were induced by E2 in control NOB cells while mutant *POC5*^A429V^ AIS osteoblasts were nonresponsive. 

The sexual dimorphism observed in AIS patients could be due to hormone deregulation. Sexual dimorphic traits are controlled by E2 in humans. In the present study, we identified POC5 as an ERα target gene regulated by E2. Other centriolar genes have also been studied in the context of estrogen. Specifically, ASPM and CDK5RAP2 (both are centriolar proteins) were found to be repressed by E2 [39]. Reporter and endogenous expression analyses showed that E2 was able to suppress the expression of these genes. The regulation of these genes by E2 was mediated by its direct interaction with predicted half-sites of ERE, but the exact mechanism of repression remains unknown [39]. Exposure to E2 was also described to upregulate Aur-A, γ-tubulin, and centrin, as well as centrosome amplification [40], suggesting that the E2 regulation of centriole proteins could confer an undetermined advantage to the cells.

In this study, the effect of *POC5* mutation on the biological activity of osteoblasts based on alizarin red indicated a lower mineralization and lower proliferation rate compared to the control and wt *POC5*-expressing cells. Low mineralization and osteopenia are known to be associated with progressive scoliosis. In AIS patients, significant osteopenia (or reduced mineral density) is clearly associated with delayed bone maturation, which represents a high-risk factor for the severity of the spinal deformity [41,42].

The estrogen insensitivity (or estrogen resistance) in AIS cells is an interesting and clinically relevant observation, as delayed puberty and lower levels of estradiol were previously reported in girls with AIS [43,44]. POC5 genetically contribute to the scoliosis predisposition and is an etiological factor in scoliosis initiation, while the suggested resistance mechanism to E2 treatment could be a factor that contributes to the curve aggravation. Most probably, in AIS cells expressing the *POC5*^A429V^ variant, the response to E2 is altered. To our knowledge, our work is the first to report the effect of E2 on *POC5* synthesis and regulation. There is little work on the role of POC5. In humans, POC5 localizes to the distal portion of centrioles, and its recruitment to procentrioles is essential for full centriolar maturation and normal cell-cycle processing. This centrosomal protein interacts with centrin and inversin [25], both involved in cell division, polarity, and motility. More recently, *POC5* variants were reported in the Chinese scoliotic population. Although this recent case-control study reported that a common variant (rs6892146) of *POC5* was associated with susceptibility to AIS [24], this study also shows significantly higher mRNA expression of *POC5* in the muscles of patients with scoliosis compared to the controls. 

There are several studies on the role of genetic factors in AIS where mutations in several genes have been associated with the development and progression of AIS [11,14,15,18,45,46,47,48,49,50,51]. However, little is known about possible targets affected by estrogen in AIS. Our study provides evidence of an altered regulation of *POC5* in AIS cells and identifies *POC5* as a gene regulated by estrogen. Interestingly, ChIP experiments showed similar E2 responses of the ERE1/2- and ERE3-containing promoter regions of the POC5 gene, thereby supporting a functional role of the proximal contig of EREs in driving the estrogenic upregulation of the POC5 gene. 

Another interesting finding was observed in this study; we found that treatment with E2 induced the amplification of POC5 expression and the formation of linear centriolar structures. Centrosomes are considered amplified in number when more than two centrosomes are found to be associated with a single nucleus. It was found that the overexpression of POC5 led to the formation of linear structures, which were dependent on centrin that serves to recruit other centrosomal proteins [26]. Several mechanisms have been described that lead to centrosome amplification including cytokinesis failure, mitotic slippage, cell-cell fusion, over-duplication of centrioles, and de novo centriole assembly. Centrosome amplification may result from centriole over-duplication, for example, through the overexpression of centriolar proteins, which most probably underlies the mechanism by which POC4 is overexpressed. This is consistent with the role of Polo Like Kinase 4 (Plk4) associated with centrioles, either in basal bodies or centrosomes [52]. Over-expression of Plk4 during development leads to increased numbers of centrosomes in the basal epidermis. These findings complement our results on POC5. Centriole numbers are usually under tight cell-cycle control in most proliferating cells. However, cells that line the epithelia of the respiratory and reproductive tracts form hundreds of centrioles in order to provide the basal bodies for the formation of beating cilia [52]. Future work is needed to focus on the impact of centriole amplification on the cell cycle in normal and AIS cells.

## 5. Conclusions

Several studies have reported that E2 promotes the onset and development of AIS. *POC5* was identified as one of the genes causing AIS. In this study, we present evidence on the role of estrogen in gene regulation with differences observed in normal human osteoblasts compared to cells derived from AIS patients. We show that estrogen plays an important role in the regulation of POC5. Knowledge of the role of POC5 in AIS is growing and developing. Based on our findings, there seems to be a differential regulation of POC5 by E2 in normal and AIS osteoblasts, suggesting a possible mechanism of estrogen resistance in AIS. 

## Figures and Tables

**Figure 1 genes-14-01111-f001:**
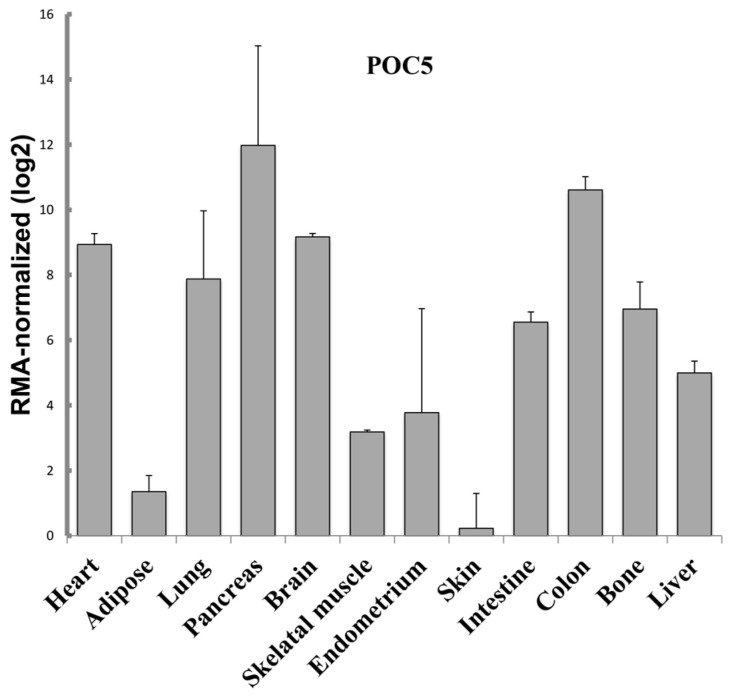
*POC5* expression in human tissues. The expression profile of *POC5* in different tissues was studied using the GEO database (GDS3834/10890). *POC5* is expressed in a wide range of tissues with high levels in the pancreas, lung, brain, heart, and colon.

**Figure 2 genes-14-01111-f002:**
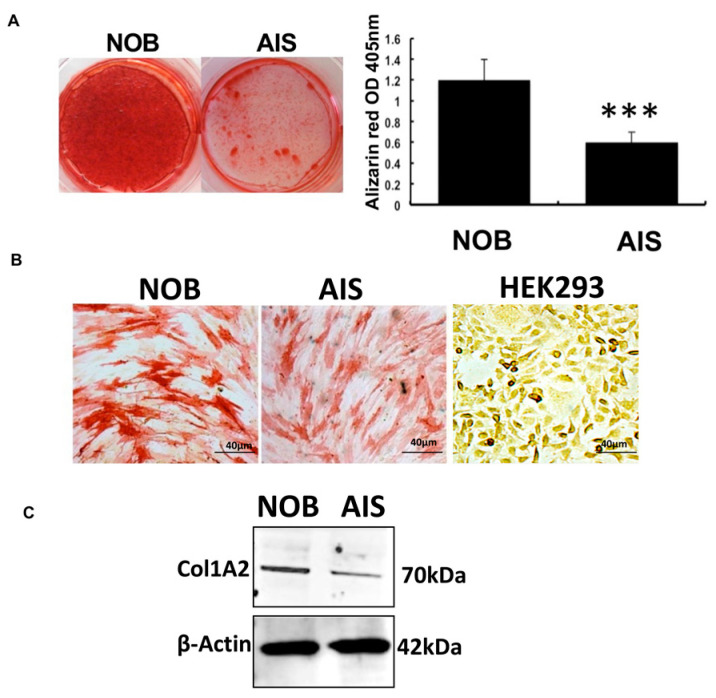
Osteoblast Characterization. (**A**) Human osteoblast (NOB and AIS) cells were cultured in osteogenic medium (ascorbic acid, glycerol phosphate, and dexamethasone) for 28 days. After 28 days, alizarin staining showed reduced mineral deposition in the AIS cells compared to NOB cells (left). Quantification of alizarin red staining (right) measured at an absorbance of 405 nm. *** *p* < 0.001. (**B**) Alkaline phosphatase activity detection in osteoblasts derived from the bone of NOB, AIS, and HEK293 cells (negative control) (Scale bar 40 μm). (**C**) COL1A2 protein expression in NOB and AIS cells. AIS cells had lower levels of COL1A2 compared to NOB cells.

**Figure 3 genes-14-01111-f003:**
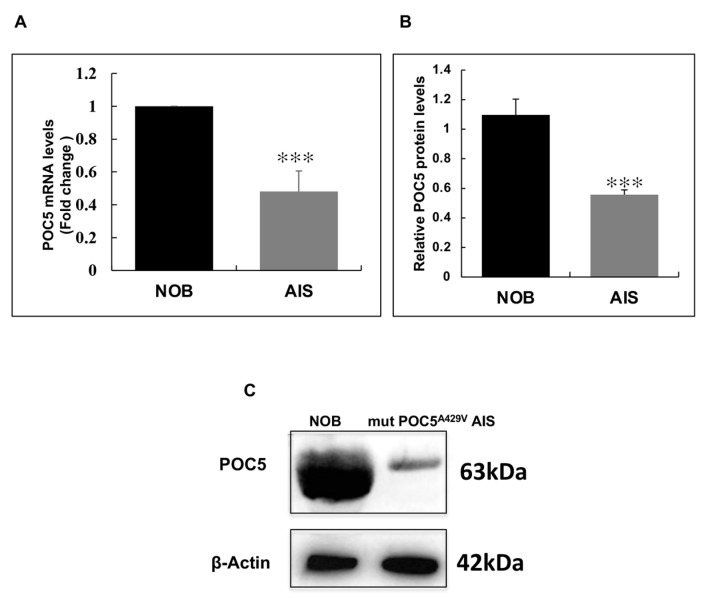
POC5 expression in human normal control (NOBs) and mutant *POC5*^A429V^ AIS osteoblasts. (**A**) qPCR of the expression of *POC5* in control and mutant *POC5*^A429V^ AIS osteoblasts. The level of expression of *POC5* was higher in control NOB cells than in mutant *POC5*^A429V^ AIS osteoblasts. *GAPDH* was used as an endogenous control. Fold change with respect to NOB cells was calculated using the delta cT method (2^−ΔCT^). For qPCR, error bars represent ± S.D of three independent experiments performed in triplicate (*p* < 0.01). *** *p* < 0.001 denotes significant differences between NOBs and mutant *POC5*^A429V^ AIS osteoblasts. Student’s *t*-test was used to determine statistical significance. (**B**) Western blot quantification shows that the endogenous protein levels of POC5 in normal osteoblasts were significantly higher than those of human mutant *POC5*^A429V^ AIS osteoblasts in the absence of E2. *** *p* < 0.001. One-way ANOVA was used for statistical analysis. (**C**) Western blot of the protein expression of the basal levels of POC5 in NOBs and mutant *POC5*^A429V^ AIS osteoblasts. β-actin was used as a loading control. Band intensity was measured using Image J (Java 13.0.6, 64-bit), and the ratio of POC5 to β-actin was calculated. The results are the mean ± SD of three independent experiments (n = 3).

**Figure 4 genes-14-01111-f004:**
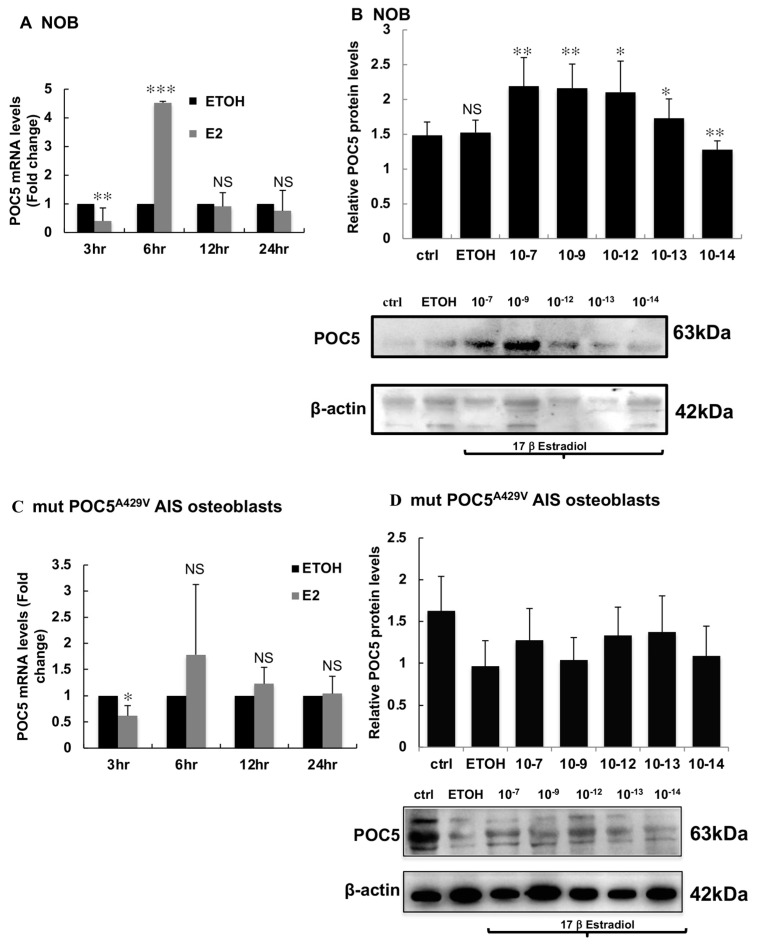
Dose-dependent regulation of POC5 by E2 in normal and mutant *POC5*^A429V^ AIS osteoblasts. (**A**) *POC5* gene expression was regulated by estradiol. qPCR of NOBs treated with 10^−7^ M E2 at different time points (3 h, 6 h, 12 h, 24 h) using specific primers for *POC5* and *GAPDH* (used as an endogenous control). Maximal response to E2 was obtained at 6 h. ** *p* < 0.01, *** *p* < 0.001. (**B**) WB was performed on NOBs treated with different doses of E2 ranging from 10^−14^ M to 10^−7^M. E2 upregulated POC5 expression in a dose-dependent manner. Maximal response was obtained between 10^−12^ and 10^−7^ M * *p* < 0.05, ** *p* < 0.01. Paired means of ETOH and E2 at different doses and the control (Ctrl) were compared with one-way ANOVA. (**C**) qPCR of mutant *POC5*^A429V^ AIS osteoblasts treated with E2 as described above. Although the response to E2 reached maximum levels after 6 h of induction, this was not significant. (**D**) WB expression of POC5 at different doses of E2 (10^−14^ M to 10^−7^ M) in mutant *POC5*^A429V^ AIS osteoblasts. No significant change was observed after treatment with different doses of E2. Bands corresponding to the correct molecular weight are indicated for both POC5 (63 kDa) and β-actin (42 kDa). For qPCR, error bars represent ± S.D of three independent experiments performed in triplicate. For quantification of band intensity, Image J was used, and the ratio of POC5 to β-actin was calculated (n = 3). The results are the mean ± SD of three independent experiments. NS: not significant. Paired means of ETOH, E2 at different doses, and the Ctrl were compared by one-way ANOVA. (**E**) Immunofluorescence to study the localization of POC5 in response to E2 using anti-POC5 and anti-centrin antibodies. POC5 cellular localization in both normal and mutant *POC5*^A429V^ AIS osteoblasts in the absence and presence of 10^−7^ M E2 centrin (green), POC5 (red), and DAPI (blue). (**F**) Bar diagram representing the POC5 dot intensity measured using Image J. *** wt+E2 represents a statistically significant increase in POC5 staining (in the linear centriolar structures). Approximately 100 cells were counted in each group. *** *p* < 0.001, NS: not significant. Scale bar 20 μm.

**Figure 5 genes-14-01111-f005:**
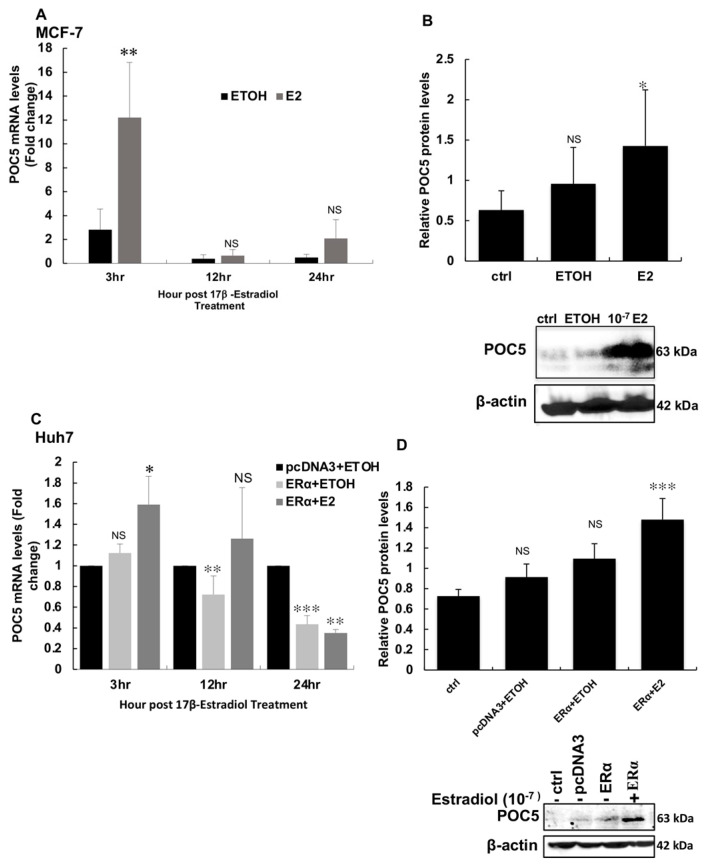
Regulation of POC5 by E2 in MCF−7 and Huh−7 cells. (**A**) qPCR analysis of MCF−7 exposed to 10^−7^ M E2 for different time points (3 h, 12 h, 24 h) using specific primers for *POC5* and *GAPDH*. Data are presented as the fold change with respect to vehicle treatment. Maximal response was obtained at 3 h post-treatment with E2. The levels returned to basal levels at 12 h and 24 h. ** *p* < 0.01, NS: not significant. Statistical significance between ETOH and E2 at three different time points (3 h, 12 h, 24 h) was compared. One-way ANOVA was used for statistical analysis. (**B**) POC5 protein expression in MCF-7 exposed and not exposed to E2. E2 (10^−7^ M) upregulated POC5 expression. * *p* < 0.05, NS: not significant. Paired means of non-treated (ctrl), ETOH, and E2 were compared. One-way ANOVA was used for statistical significance. (**C**) qPCR was performed in Huh-7 cells treated with 10^−7^ M E2 for different time points (3 h, 12 h, 24 h) as described above. At 3 h, E2 increased *POC5* expression. The expression of *POC5* was downregulated at 24 h. * *p* < 0.05, ** *p* < 0.01, *** *p* < 0.001. Student’s *t*-test was used for statistical comparisons of pcDNA3+ETOH and ERα+ETOH and ERα+E2 at three time points (3 h, 12 h, 24 h). (**D**) Protein expression of POC5 in response to 10^−7^ M E2; there was an increase in the protein levels of POC5. For qPCR, error bars represent ± S.D of three independent experiments performed in triplicate. For quantification of band intensity, Image j was used and the ratio of POC5 to β-actin was calculated (n = 3). The results are the mean ± SD of two independent experiments. *** *p* < 0.001, NS: not significant. Paired analysis of the means of ctrl non-transfected (−E2), pcDNA3-E2, ERα-E2, and ERα+E2 compared with one-way ANOVA. The arrow indicates the band corresponding to the molecular weight of POC5 (63 kDa).

**Figure 6 genes-14-01111-f006:**
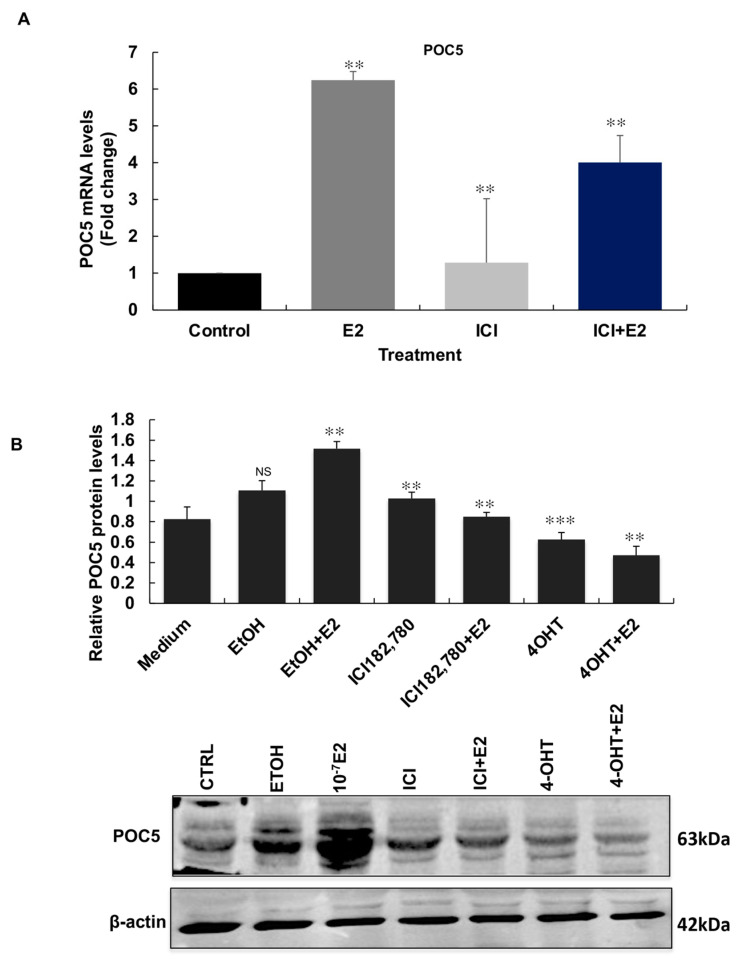
ERα antagonists reverse the up-regulatory effects of E2 on POC5 expression. (**A**) ERα antagonist ICI182, 780 significantly impaired the upregulatory effect of E2 on *POC5* mRNA levels. Error bars represent ± S.D of three independent experiments performed in triplicate. ** *p* < 0.01. The effects of different treatments (control, E2, ICI-182, 780, ICI-182, 780+E2) were analyzed by one-way ANOVA. ** E2 indicates that *POC5* expression in E2-treated cells was significantly upregulated (*p* < 0.01) compared to control cells. ** ICI 182,780 indicates a significant downregulation of *POC5* expression compared to E2-treated cells. ** ICI 182, 780+E2 indicates that POC5 expression was significantly upregulated compared to ICI 182, 780. (**B**) MCF-7 cells were treated with ETOH (0.1%), 10^−7^ M E2, or the estrogen receptor antagonists 10^−7^ M ICI-182, 780, 10^−7^ M 4-OHT and then subjected to Western blot using the POC5 antibody. β-actin was used as a loading control. ICI-182, 780 and 4-OHT treatments inhibited the E2-induced upregulation of POC5. ** *p* < 0.01, *** *p* < 0.001, NS: not significant. Statistical analysis was performed by one-way ANOVA. Paired analysis of the means of (ETOH, CTRL), (E2, CTRL), (ICI-182, 780, E2), (ICI-182, 780, ICI182, 780+E2), (4OHT, E2), and (4OHT, 4OHT+E2 was performed. For quantification of band intensity, Image J was used and the ratio of POC5 to β-actin was calculated (n = 2). The results are the mean ± SD of two independent experiments. The arrow indicates the band corresponding to the molecular weight of POC5 (63 kDa).

**Figure 7 genes-14-01111-f007:**
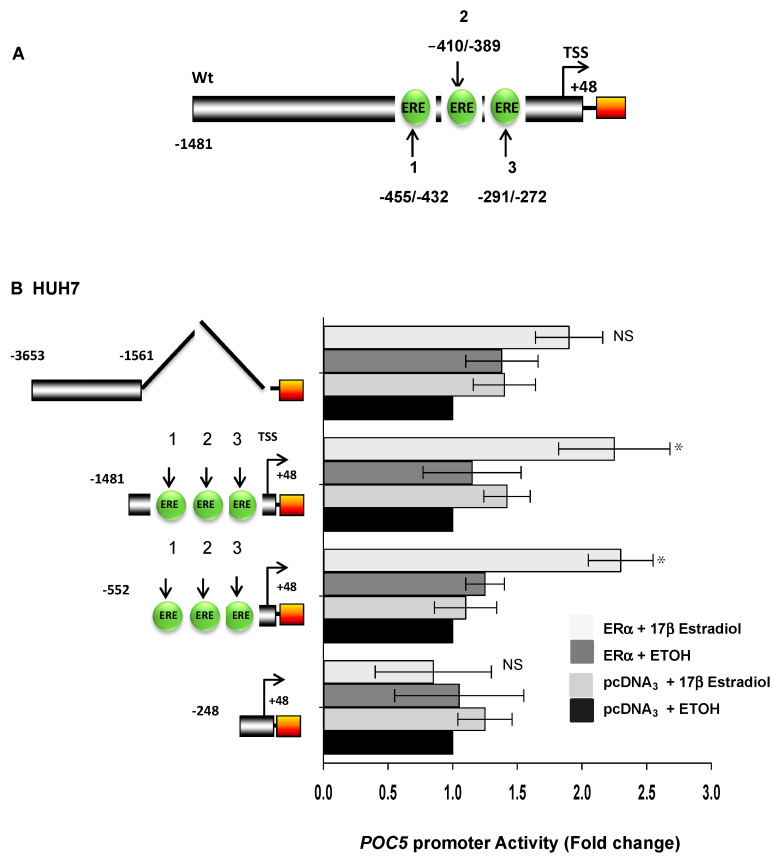
Regulation of *POC5* by E2 at the promoter level through ERα. (**A**) Schematic representation of the promoter fragment 1481/+48 with the corresponding estrogen response elements (EREs) indicated as 1, 2, and 3. (**B**) Luciferase assay was performed in Huh−7 cells using the four reporter constructs to the left of the y-axis. The promoter constructs were co-transfected with the effector vector coding for the estrogen receptor α or with an empty vector (pCDNA3). All wells were transfected with the Renilla luciferase vector for normalization. Twenty-four hours after transfection, the cells were treated with either estradiol or the vehicle (ethanol). Treatment with estradiol upregulated the POC5 promoter activity of constructs −1481/+48 and −552/+48. The data are expressed as the mean ± SD of three experiments performed in triplicate (n = 3). Means that were significantly different from the control are indicated by * *p* < 0.05; NS: not significant. The different effects of treatments were analyzed by one-way ANOVA.

**Figure 8 genes-14-01111-f008:**
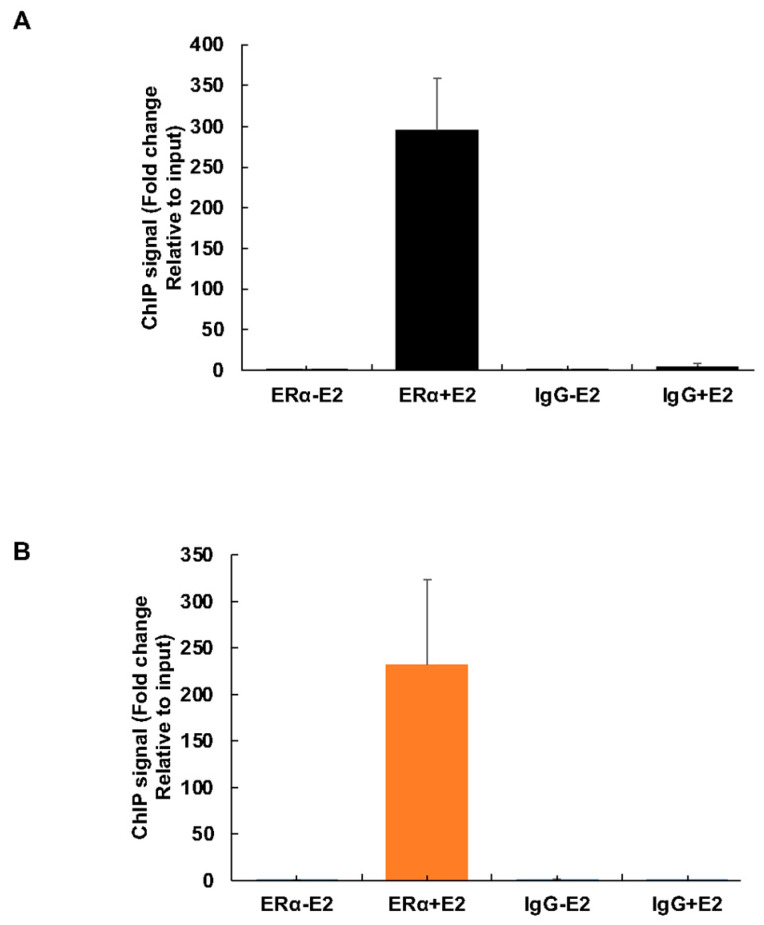
Estrogen receptor binds to the *POC5* promoter. (**A**,**B**) Chromatin was isolated from MCF-7 cells that were not treated or treated with 10^−7^ M E2 for 1 h prior to ChIP, and immunoprecipitations were performed with an antibody specific to ERα or a non-specific IgG. qPCR reactions were carried out with the various ChIP samples (ERα-E2, ERα+E2, IgG-E2, IgG+E2, Input-E2, Input+E2). Primers that amplify the region containing ERE1/2 (**A**) or ERE3 (**B**) were used (Table 4). The fold changes in the values for E2- and E2+ were calculated and graphed. Error bars represent ± S.D of three independent experiments performed in triplicate. Means of ERα-E2 and ERα+E2, IgG-E2, and IgG+E2 were compared.

**Figure 9 genes-14-01111-f009:**
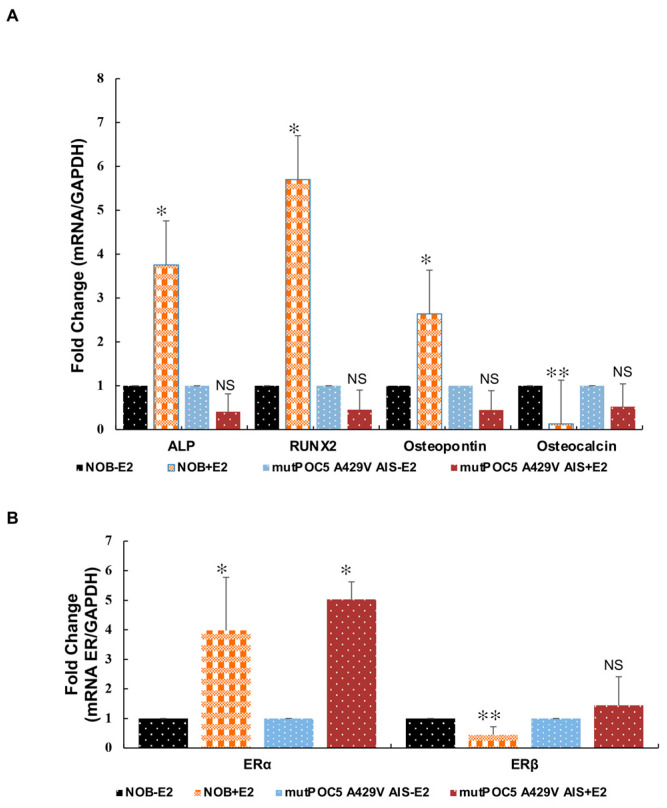
qPCR of gene expression of different markers of differentiation and ERs in osteoblasts. (**A**) RNA was extracted from NOBs and mutant *POC5*^A429V^ AIS treated with vehicle or 10^−7^ M 17-β-estradiol (E2) for 24 h. *ALP*, *RUNX2*, and *SPP1* were upregulated, and *BGLAP* was downregulated by E2 in NOB cells. No significant change was observed in mutant *POC5*^A429V^ AIS osteoblasts cells. Expression was normalized to *GAPDH* and is plotted as the fold increase relative to the vehicle-treated sample. *ALP*: alkaline phosphatase; *RUNX2*: Runt-related transcription factor 2; *SPP1*: secreted phosphoprotein1; *BGLAP:* Bone γ carboxyglutamate protein; * *p* < 0.05, ** *p* < 0.01, NS non significant. Statistical analysis was performed between +E2- and −E2-treated NOBs and mutant POC5^A429V^ AIS osteoblasts. Student’s *t*-test was used. (**B**) qPCR of expression of *ERα* and *ERβ.* NOBs and mutant *POC5*^A429V^ AIS osteoblasts were exposed to vehicle or 10^−7^ M E2 for 24 h. *ERα* expression was induced by E2 in NOBs and mutant *POC5*^A429V^ AIS osteoblasts. *ERβ* was downregulated in NOBs. Expression was normalized to *GAPDH* and is plotted as the fold increase relative to the vehicle-treated sample. Error bars represent ± S.D of three independent experiments performed in triplicate. * *p* < 0.05, ** *p* < 0.01, NS non significant. Statistical analysis was performed between +E2- and −E2-treated NOBs and mutant *POC5*^A429V^ AIS osteoblasts. Student’s *t*-test was used.

**Table 1 genes-14-01111-t001:** Clinical data of the patients. The corresponding sex, age, Cobb angle, and curve type are indicated for each patient. Rt: right thoracic, Rtl: right thoracic lumbar, ll: left lumbar.

Sex	Age of Surgery	Cobb Angle	Curve Type
F	14	99–66	rt/ll
F	12	54	rt/ll
M	13	75	rt
F	14	68	rt
F	11	54–37	rt
F	18	37–52	rt/ll
F	13	56	rt
F	14	67	rt
F	16	64	rt
F	12	68	rt
F	16	58	rt
F	13	66	rt
F	13	51	rt
F	14	78	rt
F	14	41–48	rt/ll
F	13	54	rt
F	14	88	rt
F	15	32	rtl
F	18	50	rtl
F	13	58–49	rt
F	16	53	rt/ll
F	12	74–62	rt
F	14	41–50	rt/ll
F	14	81–59	rt/ll
F	13	53	rt
F	15	55–42	rt/ll
F	13	61	rt
F	15	58	rt
M	14	61	rt
F	15	72–59	rt/ll
F	16	28	rt
F	12	65	rt
F	15	42	rt
M	18	70	rt

**Table 2 genes-14-01111-t002:** Primers used for qPCR.

Primer Name	Primer Sequence
*GAPDH*_Sense	5′AGGAGTAAGACCCCTGGACC3′
*GAPDH*_AntiSense	5′GGAGATTCAGTGTGGTGGGG3′
*POC5*_Sense	5′CATGTCAGAGCCAGACAGGA3′
*POC5*_AntiSense	5′GGAACGCCAGACTTTCCAGA3′
*ALP*_Sense	5′ACACCTGGAAGAGCTTCAAACCGA3′
*ALP*_AntiSense	5′TCCACCAAATGTGAAGACGTGGGA3′
*OSTEOCALCIN*_Sense	5′ACACTCCTCGCCCTATTG3′
*OSTEOCALCIN*_AntiSense	5′GATGTGGTCAGCCAACTC3′
*RUNX2*_Sense	5′TCCGGAATGCCTCTGCTGTTATGA3′
*RUNX2*_AntiSense	5′ACTGAGGCGGTCAGAGAACAAACT3′
*OSTEOPONTIN*_Sense	5′CAGCCATGAATTTCACAGCC3′
*OSTEOPONTIN*_AntiSense	5′GGGAGTTTCCATGAA GCCAC3′

**Table 3 genes-14-01111-t003:** List of primers used to clone the *POC5* promoter full-length and deletion constructs. The underlined nucleotide sequence represents the restriction site that was introduced for cloning. Primers are shown as sense and anti-sense, representing the forward and reverse directions, respectively. The second column indicates the beginning and end positions of each primer sequence with respect to the transcriptional start site.

Primer	Position	Sequence
Sense	−3661/−3641	5′CCAGCAGGCTAGCCCAGGCGT 3′
Anti-sense	−1478/−1448	5′GTCTTTCAACTTACATTGGCAACAGATAGGC 3′
Sense	−1481/−1455	5′CATACCTGCTAGCCTATCTGTTGCCAATGTAAGTT 3′
Sense	−555/−536	5′CTGCACACCAGCCTGGACGGGCTAGCAAGACTCCATCTCAAAA 3′
Sense	−248/−230	5′GAAAGCCAACAGCACACGGCGCTAGCCAACTTCAGCCCTGC3′

**Table 4 genes-14-01111-t004:** List of primers used for chromatin immunoprecipitation (ChIP) of the ESRα response element on the *POC5* promoter.

Primer Name	Sequence
POC5_CHIP_ERE1/2_Sense	5′ GGGATCTGTGGATGATGCAG 3′
POC5_CHIP_ERE1/2_Antisense	5′ GAGTTCGAGACTAGTCTGGG 3′
POC5_CHIP_ERE3_Sense	5′ GCCCAGAATTTCGGATTGTTC 3′
POC5_CHIP_ERE3_AntiSense	5′ CTGAAGTTGGCTAATGCCGTG 3′

## Data Availability

Data used or analyzed during the current study are available from the corresponding author upon reasonable request.

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
