# Peer review of "Differential Regulation of POC5 by ERα in Human Normal and Scoliotic Cells"

_genes, 2023, doi:10.3390/genes14051111_

Round 1

Reviewer 1 Report

The manuscript by Hassan et al. comprehensively analyzed the role of POC5 in normal osteoblasts (NOB) and other ER-positive cells. The study identifies POC5 as a gene regulated by estrogen, which was essential to increase the expression of bone markers. The major findings and conclusions are convincing. The study provides valuable resources to understand the pathogenic mechanism of adolescent idiopathic scoliosis. Following are the points of criticism.

Major points:
1. In Figure 7B, are these three putative estrogen response elements (ERE) functionally redundant? Similarly, in Figure 8B, are the ERE2 and ERE3 also enriched in the ChIP?

2. In Figure 9B, the expression of ER
α increased in NOB when treated by E2. Why did the expression ERβ decrease, in NOB+E2 group compared to the control, NOB-E2?

Minor points:

1.     Typos in the manuscript. In the title, “Differential redulation of  POC5 by ERin normal and Hu-2 man Scoliotic Cells”, there are two blanks and one typo, “redulation”.

2.     In Figure 1, what is the GEO access number of the database? It should be indicated in the method and figure legend.

3.     In Figure 2B, the scale bar was missing.

4.     In Figure 2C, the NOB and AIS cells should be indicated in the figure.

5.     In Figure 2C, 3C, 4B, 4D, 5D and 6B, the original western blot gel image without cropping should be provided in the supplementary material.

6.     In Line 293, the legend of Figure 3C was missing.

There are several typos in the manuscript, which need proofreading.

Reviewer 2 Report

In this manuscript, the authors identify POC5 as an estrogen-responsive gene under the regulation of estrogen receptor ER in NOB and other ERα positive cells. Using promoter activity, gene and protein expression assays, authors found that the POC5 gene was upregulated by treatment of osteoblasts with estradiol (E2) through direct genomic signaling. They observed different effects of E2 in NOB and mutant POC5A429V AIS osteoblasts. Using promoter assays, they identified an estrogen response element (ERE) in the proximal promoter of POC5 which conferred estrogen responsiveness through ER. Recruitment of ERα to the ERE of the POC5 promoter was also potentiated by estrogen. Authors’ findings suggest that estrogen is an etiological factor in scoliosis through deregulation of POC5. Based on my evaluation, I believe that this work has the potential to make a significant contribution to the field.

To make the manuscript more profound, we suggest adding the following modifications:

1.     In the Discussion section, please discuss other genes that have been identified as causative in AIS. This will provide a broader context for your findings and help readers understand how POC5 fits into the larger picture of AIS etiology.

2.     To further support your conclusion that ERα can bind to the promoter of POC5, we suggest performing Electrophoretic mobility shift assay (EMSA). This will provide more direct evidence of protein-DNA interactions.

3.     Regarding Figure 2B, the band for β-actin binding is not clear. We recommend redoing the Western blot to improve the quality of the image.

English language quality is good.

Reviewer 3 Report

Summary

This manuscript clearly reported the estrogen (using 17β-estradiol [E2] in the experiments)-dependent expression of protein of centriole 5 (POC5) in osteoblasts. E2 drives estrogen receptor α (ERα) to estrogen regulatory elements (ERE) in the POC5 promoter to induce POC5 expression. This mechanism is impaired in osteoblasts with the POC5-A429V mutation, resulting in reduced osteogenic gene expression and osteoblast mineralization. It is suggested to be one of the mechanism of adolescent idiopathic scoliosis (AIS) caused by POC5 mutation. This study provides novel insights into the pathogenesis of AIS.

The results are clear and convincing. A few minor changes, as noted below, will make the manuscript perfect. The manuscript should be professionally proofread because of some careless mistakes (e.g. upper/lower case).

Comments

1.          Line 93: Huh-7 cells should be briefly explained as MCF-7 was (Line 94).

2.          Table 1: Please explain the meaning of the two numbers in the Age row.

3.          Section 2.5: The title of the section should be “Alkaline phosphatase (ALP) staining”.

4.          Figure 1: Which datasets did the authors used? An accession number or some other information should be provided for readers to access the original data.

5.          Figure 4E: The magnification of the images differ between upper and lower panels. Scale bars should also be displayed in the upper panels.

6.          Figure 4E: Images without E2 should be provided as they were quantified in Figure 4F.

7.          Figure 7B: The order of the bars is very confusing. It should be reversed.

The manuscript should be professionally proofread because of some careless mistakes (e.g. upper/lower case).

Round 2

Reviewer 1 Report

In the revised manuscript, the authors have addressed all my concerns. It can be published as it.